# Polymer Translocation and Nanopore Sequencing: A Review of Advances and Challenges

**DOI:** 10.3390/ijms24076153

**Published:** 2023-03-24

**Authors:** Swarn Lata Singh, Keerti Chauhan, Atul S. Bharadwaj, Vimal Kishore, Peter Laux, Andreas Luch, Ajay Vikram Singh

**Affiliations:** 1Department of Physics, Mahila Mahavidyalaya (MMV), Banaras Hindu University, Varanasi 221005, UP, India; 2Department of Physics, Banaras Hindu University, Varanasi 221005, UP, India; 3Department of Physics, CMP Degree College, University of Allahabad, Prayagraj 211002, UP, India; 4Department of Chemical and Product Safety, German Federal Institute of Risk Assessment (BfR) Maxdohrnstrasse 8-10, 10589 Berlin, Germany

**Keywords:** polymer translocation, nanopore sequencing, translocation dynamics, nanopores

## Abstract

Various biological processes involve the translocation of macromolecules across nanopores; these pores are basically protein channels embedded in membranes. Understanding the mechanism of translocation is crucial to a range of technological applications, including DNA sequencing, single molecule detection, and controlled drug delivery. In this spirit, numerous efforts have been made to develop polymer translocation-based sequencing devices, these efforts include findings and insights from theoretical modeling, simulations, and experimental studies. As much as the past and ongoing studies have added to the knowledge, the practical realization of low-cost, high-throughput sequencing devices, however, has still not been realized. There are challenges, the foremost of which is controlling the speed of translocation at the single monomer level, which remain to be addressed in order to use polymer translocation-based methods for sensing applications. In this article, we review the recent studies aimed at developing control over the dynamics of polymer translocation through nanopores.

## 1. Introduction

Polymer translocation through nanopores is a process that is ubiquitous in biology (Figure 1a is a cartoon representation of polymer translocation through a nanopore). Examples include processes ranging from the transport of DNA, RNA, and protein molecules through cell membranes and nuclear pores, the packaging of the viral genome into capsid, to the ejection of viral DNA into host cells [1,2,3,4,5,6,7]. The huge interest, however, in understanding the physical mechanism behind polymer translocation is driven by its potential technological applications. Since the seminal work of Kasianowicz et al., demonstrating the possibilities of translocating DNA electrophoretically through a nanopore [8], there have been continuous efforts to develop cheap and efficient polymer translocation-based sequencing devices [9,10,11]. The underlying idea of such sequencers is as follows: When a polymer, under the influence of an external field, translocates through a nano-sized pore that provides a sole pathway to an ionic current, the ionic current through the pore changes. Each passing monomer gives a characteristic variation to this ionic current and recording of the current gives a map of the underlying polymer structure (see Figure 1b which is a cartoon representation of a polymer passing through a nanopore, giving rise to current modulations). Though there are other technological applications based on polymer translocation, e.g., controlled drug delivery [12,13,14], separation of polyelectrolytes, and filtration of macro-molecules [15,16], the biggest motivation behind studying polymer translocation lies in its applications in sequencing devices.

The information encoded in DNA sequences is of paramount importance in gaining an understanding of various disease mechanisms and in improving genetic diagnoses [17,18]. In principle, polymer translocation-based sequencing is a straightforward process; an ssDNA passing through the nanopore in a head-to-tail fashion (one nucleotide at a time) gives rise to modulations in the ionic current readout, and these modulations correspond to the underlying sequence. In practice, however, this has not yet been realized, despite tremendous and continuous efforts. One of the key challenges is the speed of the translocation; often, on average, one nucleotide traverses a nanopore in less than one micro-second [19,20] for solid-state nanopores. This speed makes it difficult to detect the signal difference between the individual nucleotides [9,21]. Consequently, a wealth of research has been carried out on decelerating the translocation process; an overview of these studies is the focus of the present manuscript. There are other issues apart from the translocation speed, such as the low capture rate [22,23,24], thermal fluctuations [25], and achieving/engineering the right kind of nanopore [9].

The first and foremost element of *developing control over translocation dynamics* is to develop an understanding of polymer translocation, which actually is a complex process. There are three basic scenarios for polymer translocation: (i) the translocation takes place in the absence of a net driving force and is known as *unforced or unbiased translocation*, (ii) the translocation happens under the influence of a net force gradient across the channel, called *forced or driven translocation*, or (iii) a more recent approach, known as *pulled polymer translocation,* in which the polymer is pulled through the nanopore using atomic force microscopy or optical/magnetic tweezers. We will discuss the basics of polymer translocation in Section 2, focusing more on driven and end-pulled translocation as these are technologically more important. Moreover, there is a combined effect of various parameters such as the properties of the solvent, the pore geometry, pore–polymer interactions, and external forces (if any) that affect the translocation process. These parameters can thus be used to tune the translocation dynamics. We will elaborate more on how the above-mentioned parameters can be used to manipulate the translocation process in Section 4.

**Figure 1 ijms-24-06153-f001:**
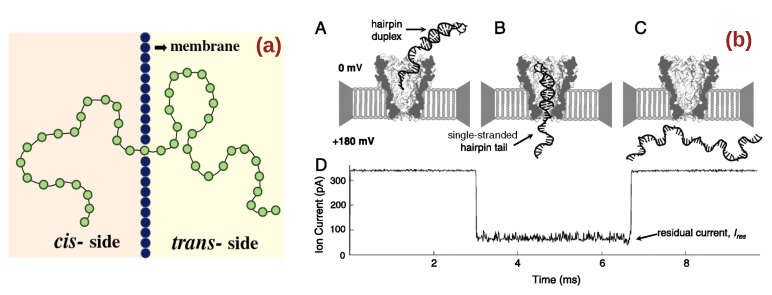
(**a**) Schematics of the configuration of a translocating polymer through nanopore in a membrane. (**b**) A cartoon representation of various stages of DNA translocation through an MspA pore (A–C) and the corresponding residual current in time in (D), reproduced from [26].

Having a suitable nanopore is another crucial parameter for obtaining good resolution during sequencing. As an example, if ssDNA has to be sequenced and the pore is wide enough to allow a hairpin translocation, the sequencing would not be reliable. Similarly, if the width of the pore is of the order of thickness of ssDNA, it cannot be used to sequence a double stranded DNA (dsDNA). The thickness of the membrane plays a role too; if the membrane is much thicker than the distance between two nucleotides, detecting the signal at a single nucleotide level would be difficult. In Section 3, we will discuss the nanopores briefly. Nanopores could be of biological origin, could be synthesized using thin films of inorganic material (known as solid-state nanopore or SSNPs), or a hybrid [10] of both the biological and the solid-state nanopores. We will list the merits and limitations of each of these three. In Section 4, we will give an account of the studies focusing on slowing down the rapid translocation of DNA during sequencing. Our main goal here is to present all the efforts and outcomes of thereof regarding the manipulation of polymer transport dynamics in one place. We end the article with a short summary and outlook in Section 5.

## 2. Polymer Translocation: Highlights from Theoretical Development

Translocation consists of three distinct phases; first, the polymer has to reach the vicinity of the pore. The balance of the forces acting on the polymer (diffusion, electro-chemical gradient across the pore, and external forces, if applied) defines a capture radius for the polymer. Once within the capture radius, one of the ends of the polymer must find the pore entrance, which requires crossing a considerable entropic barrier. The exact value of the barrier depends upon system parameters. While entering the pore, the polymer again has to overcome conformational entropy. Afterward, the polymer transits the pore through the drift-diffusion process. Theoretical formulation of polymer translocation is a difficult task as it is a multi-step, highly non-equilibrium process dictated by a combination of a number of forces. The most relevant quantity while studying translocation is the translocation time (τ) and its scaling with the polymer length. Generally, translocation can be divided into two categories: in absence of any external force (unbiased) and in the presence of an external force. The latter can be implemented in two ways, either by setting up a force gradient across the pore or by applying a pulling force on one end of the polymer. In the following, we will briefly discuss the theoretical/computational studies for each case.

The first important theoretical model for unforced translocation was proposed by Peskin et al. in 1993 [27], where they showed that the random thermal motion of the polymer, combined with chemical asymmetry across the pore, creates directional diffusion. This was followed by the Monte Carlo study by Baumgärtner and Skolnick in 1995 [28], who studied a curved membrane and the translocation was driven by curvature-induced membrane fluctuation. The random thermal fluctuations are modified into directional diffusion in the presence of any asymmetry across the pore, known as Brownian ratcheting. Ratchet theory is a possible explanation for many fundamental biological operations. The importance of these papers is that they made remarks on the diffusive-directed motion of the polymer during the translocation, a model similar to the Huxley model for myosin [29]. The analytic, quantitative theory proposed by Sung and Park [30] on the basis of statistical physics of polymer and stochastic processes focuses on calculating the key features of polymer translocation, e.g., the free energy barrier and mean first passage time (translocation time). Sung and Park also assumed the diffusion coefficient of the chain to be a function of the chain length; $D\propto N^{-1}$ for a Rouse chain and $D\propto N^{-0.5}$ for a Zimm chain. The mean translocation time was found to follow the scaling $\tau \propto N^{\alpha }$, where α=3 and 2.5 for Rouse and Zimm chains, respectively. Mutthukumar [31] corrected this by assuming the diffusion coefficient to be a function of the local details of the pore–monomer interaction and independent of the chain length. They obtained the scaling behavior of $\tau \propto N^{2}$ for unbiased translocation. These early studies, however, assumed that the chain threads so slowly that the process could be considered a quasi-equilibrium process. This turned out to be an over-simplification of the problem in the scaling limit, as was first pointed out by Chuang et al. [32]. The time taken by a freely diffusing Rouse polymer to move a distance of its own size scales with $N^{1+2\nu }$, where ν is the Flory exponent, and this exponent is always greater than two in the scaling limit. Chuang et al. argued that translocation through a nanopore should always be slower than the diffusion of the free polymer due to geometric constraints, and the exponent for free diffusion should set the lower bound for α in the case of translocation. They further used a computer simulation to study the unbiased translocation of a polymer using the bond fluctuation model and concluded that τ scales as $N^{1+2\nu }$. An immediate consequence of this scaling is that the translocation dynamics are anomalous; the mean square displacement of the monomer inside the pore does not vary linearly with time but follows a power law. This was followed by a number of studies reporting the scaling exponent for unbiased translocation; however, there is not a common consensus yet [33,34,35,36,37,38,39,40,41,42,43,44,45]. There is also a rich literature regarding the effect of various system parameters such as viscosity, salt concentration, chaperons, the presence of crowders, the activity of these crowders, etc., on the dynamics of unbiased translocation [46,47,48,49,50,51,52,53,54].

The other scenario, which is technologically more important, is translocation under the influence of an external field. The field can be applied either across the pore or on the head monomer; the latter approach is known as pulled translocation. In the quasi-equilibrium approach, the external field enters the equation of motion as an additional drift term giving rise to a constant drift, on top of the diffusive motion, towards the trans side. This drift velocity is proportional to the applied field, implying that the translocation time will be inversely proportional to the applied field strength. The scaling of the mean translocation time for forced translocation is thus connected to both the polymer length and the applied field ($\tau \propto N^{\alpha }/f^{\delta }$). If the driving force is very weak, entropy still plays a more significant role and the scaling corresponds to that of unbiased translocation, whereas for a strongly driven regime, the driving force takes over. Again, this oversimplified picture of the quasi-equilibrium assumption was challenged by Kantor and Kardar [55], following the same line of arguments as in the case of unbiased translocation. They concluded that the lower bound for α, in this case, is 1+ν and that the translocation is an anomalous process. Again, a rather large number of studies followed, either confirming or contesting the scaling exponent [56,57,58,59,60,61,62,63,64,65,66,67,68,69,70]. A large variation in the value of the exponent also suggests that the results are very sensitive to details such as the size of the pore, the pore polymer–interaction, the various lengths involved, etc. Apart from an externally applied field, the driving force can also be derived from other sources such as confinement [26,71,72,73,74,75] or the presence of chaperons/crowders [53,76,77,78,79,80].

The other class of forced translocation that offers better control over polymer dynamics is end-pulled polymer translocation. In this case, the force is applied on the head monomer either through an atomic force microscope or by using optical/magnetic tweezers (discussed in more details in Section 4). The most common procedure involves attaching an optical or magnetic bead to one end of the polymer and manipulating the motion of the bead using optical or magnetic tweezers. The first theoretical study of pulled polymer translocation was performed by Kantor and Kardar [55], predicting α=2 for moderate pulling forces. This was followed by a number of theoretical and numerical studies reporting the scaling behavior under different force regimes as well as different states of polymer [81,82,83,84,85,86]. The scaling exponents for various models, for both unbiased and driven translocation, are tabulated by Palyulin et al. in their review article [21].

Though most of the studies on polymer translocation consider a self-avoiding polymer in a good solvent, the conformational changes in the polymer chain during the translocation play an important role. When the polymer is in a bad solvent or the temperature is below the Θ-temperature (the temperature at which a transition from the coil to the globule state takes place), the polymer is in a globule state. He et al. studied the translocation of polymers in a bad solvent and concluded that the translocation speed increases when the quality of solvent worsens [87]. Li et al. reported that the translocation time increased for poorer solvents [88], whereas some studies have shown a non-monotonic dependence of the translocation kinetics on the solvent quality [89,90]. The scaling behavior and values of scaling exponents remain controversial as well; Yang and co-workers conducted a 3D explicit solvent dissipative particle dynamics (DPD) simulation for polymer translocation and concluded that $\tau \propto N^{\alpha }$ type scaling only holds for good solvents, and the scaling exponents could not be calculated accurately for poor solvent conditions [91]. Moisio et al. made similar observations while studying driven translocation in bad solvents using stochastic rotation dynamics coupled with molecular dynamics; a clear dependence of translocation time with polymer length was not found [92]. Li et al., however, found $\tau \propto N^{\beta }$ type scaling and calculated the values of α for Θ solvents and poor solvent conditions. In other studies, where the polymer was taken to be in the globular state either by lowering the temperature [89,93] or by tuning the interaction between the monomers (which does not exactly mimic a bad solvent situation) [94,95], the scaling of $\tau \propto N^{\alpha }$ holds. The value of α again depends upon system details such as the temperature, the magnitude of the driving force, the length of the polymer, and the strength of monomer–monomer attraction. The translocation dynamics, thus, can be controlled by varying the said parameters or even by setting an asymmetry, such as by having a good solvent on the *cis* and a poor solvent on the trans side [96].

Driven polymer translocation is fundamentally a non-equilibrium process, especially in a strong force regime, due to the long relaxation times of the polymer [62,97]. The first truly nonequilibrium treatment for polymer translocation was proposed by Sakaue [59]; the authors argued that when a polymer threads through a pore, the conformational changes are not felt by the whole chain at once [81]. Rather, the subchain on the *cis* side can be divided into two parts: the first part, closer to the pore, experiences tension due to the driving force and is mobile, whereas the second part, far from the pore, is in equilibrium. As the polymer threads through the pore, the mobile part on the *cis* side grows as the tension propagates along the backbone of the polymer. The idea was further developed into what is now known as *Iso-flux tension propagation* theory (IFTP) [98,99,100,101,102,103]. Within IFTP theory, the translocation dynamics are essentially controlled by effective friction, which includes the friction due to the pore as well as the drag on the chain. For a fully-flexible chain, only *cis*-side drag is substantial and the *trans*-side drags can be ignored. For semiflexible chains, however, the *trans*-side drag makes a contribution [103]. IFTP thus results in the equation of motion for translocation co-ordinates in terms of the effective friction, which, within some limits, could yield an exact analytic solution to the scaling of translocation time as a function of polymer length. The theory has also been extended to study end-pulled translocation [84,104] and end-pulled charged polymers [105]. The theory has also been used to study translocation in other scenarios such as translocation under an alternate driving force due to flickering pores [102], translocation into a channel [106], or even in the case of unforced translocation in the presence of active rods on the trans side [80], where the crowding results in a time-dependent net force that facilitates translocation. IFTP theory has been benchmarked against simulations and has been successful in explaining experimental results. To understand more about the theory and its applications in various scenarios, kindly refer to [84,107]. Recently, Chen et al. performed a drive translocation experiment and were able to measure the two-step, non-monotonic, non-constant velocity profile of the translocating nano-structured DNA. They observed translocation dynamics that were in agreement with IFTP theory [108].

## 3. Nanopores: A Brief Overview

Suitable nanopores are crucial to achieve high control and precision in polymer translocation-based sequencing. The initial experiments of polymer translocation were performed using biological nanopores [8,109,110,111], which typically consisted of trans-membrane proteins harvested from living cells (see Figure 1b and Figure 2b). The α-hemolysin, which is a toxin secreted by the bacterium Staphylococcus aureus and is comparatively more stable than other protein nano-channels, remains one of the most widely used biological nanopores (see Figure 2b). It inserts into a membrane to give a cylindrical nanopore. An interesting feature of α-hemolysin is the dependence of the translocation speed on the orientation of the single-stranded DNA molecule; a molecule that enters the pore from the $3^{\prime }$ end translocates slower than those entering the pore from the $5^{\prime }$ end [112]. However, the cylindrical barrel of α-hemolysin, which is 5 nm long, is capable of accommodating 10–15 nucleotides at a time, and the current readout contains the contributions from all these nucleotides [113], which affects the accuracy of detection. An alternative was found in MspA porin, which has a conical shape (see Figure 1b) with a narrow constriction at the end [26]. This narrow part allows only a few more nucleotides around the one that is translocating, and thus avoids dilution of the current readout. Many further studies have been carried out in order to tune the charge distribution (and thus the physiochemical properties) of the MspA pores through mutagenesis. The Phi29 protein channel is another attractive alternative due to its high chemical stability and possibility for mutational changes [24,114]. For recent advances in biological nanopores, kindly see [114,115]. Despite their unique advantages, such as an atomically precise structure, excellent chemical specificity, and the possibility of chemical modification through mutations, biological nanopores have some serious drawbacks, one of which is their chemical and mechanical stability. Another disadvantage of biological pores is their small diameter, which only allows biopolymers in their extended state. For example, α-hemolysin or MspA can only be used to read a single-stranded DNA or RNA molecule. Phi29 is wider in diameter and can be used to sequence double-stranded DNA. The lack of possibilities for structural manipulation is another drawback of biological nanopores.

The above-mentioned disadvantages of biological nanopores led to the development of solid-state nanopores (SSNPs) [116], which are essential pores crafted in solid-state membranes (see Figure 2a). SSNPs come with reasonable durability, stability, and control over their shape and size. The core issues involving SSNPs include (i) creating nano-sized pores of the required shape with precision and (ii) creating membranes that are thin enough to distinguish between successive nucleotides. Different groups have adopted different methods for producing nanopores of the desired shape and size. The most successful methods at the sub-nanometer scale include focused electron/ion beam sculpting [117] and controlled dielectric breakdown (CDB) [118,119]. The focused ion beam method is most commonly used for creating sub-nm pores, as the method comes with a high degree of control over the shape and the location of the pore by controlling the spot size, the dwell time, and the position of the beam on the membrane. This method, however, only works at a very high vacuum and is expensive with low throughput; hence, it is not suitable for mass production. CDB poses a promising alternative, where a voltage-induced dielectric breakdown of a thin membrane produces single nanopores of dimensions as small as 1 nm. CDB requires a simple set-up of an electric circuit that is used for applying a voltage across an insulating membrane. For more about fabrication methods and recent advances, kindly see [9,120,121,122]. Producing membranes that are thin enough to distinguish between the signal from two subsequent monomers is another huge task and an important one for high spatial resolution. First, an atomically thin membrane was prepared using a free-standing graphene sheet [123]. Single-layer membranes, however, are challenging to realize, and controlling their surface properties is a difficult task. Other methods to produce atomically thin membranes include techniques such as the thinning of the existing solid-state membranes by dry-etching or laser-assisted etching [124,125] or atomic layer deposition [126]. For a detailed study, kindly see the reviews [121,127].

Apart from the fact that creating an ultra-small nanopore in an ultra-thin membrane is challenging and the speed of translocation is very high, SSNPs also lack the chemical specificity of biological nanopores; this means that different nucleotides would interact with the nanopore in a similar manner, which is undesirable during sequencing. In order to include the chemical specificity, the surface of the nanopore could be chemically modified or functionalized [128,129]. Controlled chemical modification, however, presents its own challenges. Besides, with such modifications the precision of biological nanopores is unreachable. Alternatively, a biological nanopore could be inserted into a solid-state nanopore, resulting in what is called a *hybrid nanopore*. Such an arrangement leads to SSNPs that exhibit the precision and the complex gating behavior of biological nanopores [130]. Apart from ready proteins, DNA nano-constructs can also be used to create hybrid pores [131,132], an example of this is shown in Figure 2c. With advancing DNA nanotechnology, it is possible to construct complex DNA objects of various shapes and sizes with precise control [133,134,135]. The possibility to engineer the properties of these nano-constructs with excellent control makes these structures a promising candidate for functionalizing SSNPs to achieve the desired gating behavior. For a detailed view of the state-of-the-art in this area, kindly go through the refs. [10,132,136].

## 4. Controlling the Speed of Translocation

As mentioned earlier, the foremost challenge in sequencing a polymer at the level of individual monomers accurately is the rapid rate of translocation. In order to obtain reliable sequencing, this speed has been slowed down by approximately three orders of magnitude (from 1 µsec/nt. to 1 ms/nt.). A number of strategies have been proposed to remedy this problem, such as modifying the solvent properties [137,138], changing the topography of the nanopore [139,140,141,142,143], modifying the properties of the polymer [144,145], creating a gradient either in temperature or concentration [124,146], modifying the polymer–pore interaction [147,148], and so on. In the following, we will discuss these ideas and the work (mostly recent) undertaken around these ideas in more detail. However, first, we will discuss a rather recent approach where one end of the polymer is subjected to a pulling force using either atomic force microscopy or by using optical or magnetic tweezers. This method renders better control over the translocation process [149] and can reduce the translocation time substantially [55].

Keyser et al. demonstrated that the electrical force acting on a single DNA molecule inside a solid nanopore could be calculated using an optical tweezer. In their setup (similar to the one shown in Figure 3a, they used a laser beam to hold a DNA-grafted bead near the pore, which was immersed in a saline solution [131,149,150,151]. When a voltage was applied across the nanopore, the charged DNA molecule moved toward the pore. The opposing force, exerted by the tweezer on the optical bead, could slow down or even halt the translocation of the molecule through the pore. The same *DNA-on-bead* approach was adopted by Peng and Ling [152], but instead of using an optical tweezer, they used a magnetic tweezer to manipulate the DNA translocation. The basic idea is shown in Figure 3, though the setup is for a DNA–protein complex. The polymer to be sequenced could be grafted on an optical or magnetic bead and the translocation dynamics could be controlled by tuning the position of the bead with the help of an optical or magnetic tweezer, respectively. The sequence is reflected in the force or current vs. time curve, such as the one shown in Figure 3b. This method, since then, has been used to control the translocation of single DNA molecules, to detect/characterize/study DNA–protein complexes [153,154,155,156,157,158], and to calculate the charge of RNA molecules [159].

On the computational/theoretical front, we have mentioned the studies undertaken to reveal the translocational behavior of the pulled polymer in Section 2. The dynamics of an end-pulled polymer are different than those of a driven polymer [69,85,95,160,161,162]. Fiasconaro and Falo [163] recorded the force trajectory of an end-pulled polymer (shown in Figure 4), the analysis of which renders an estimate of the limit force to allow the translocation of each monomer in the chain. They further studied the effect of the flexibility of the pore on the translocation of an end-pulled polymer, and a resonant-like behavior was observed as a function of the longitudinal elastic parameter of the pore [164]. Chen et al. [95] studied the pulled translocation of a compact globule and it was shown that the translocation could be controlled by changing the monomer–monomer interaction. Tilahun et al. simulated the pulled translocation of a star polymer and studied the dependence of τ on the chain functionality and mass, as well as on the magnitude of the pulling force [86]. Another approach, based on the same principle as pulled translocation but less explored, involves immobilizing the DNA strand by tethering it to a bead or a probe. The motion of the DNA can thereafter be controlled by using an actuator or optical potential [165,166,167]. In Figure 5, we show the scheme of immobilizing DNA strands on a Si probe, proposed by Akahori et al. [165]. The motion of this probe was controlled using a piezo-actuator and stepper motor.

Another valuable strategy involves *working on the pores*, which includes modifying the shape and size of the nanopores, altering the surface properties, modifying the pore–polymer interaction, and so on. We will go through these strategies in the following, starting with narrowing down the pores. Reducing the width of the pores so much that only one monomer is allowed at a time slows down the speed of the translocation [124,168,169,170,171,172,173]. Narrow pores also help to avoid a hairpin translocation, which is desirable for sequencing purposes. Ultra-small nanopores, however, come with their own issues, such as reduced capture probability and a broad distribution of dwell time [23,174]. Pores in the membranes that are thinner than the distance between two monomers are supposed to have a higher spatial resolution, as the contributions from the nearby nucleotides to the current readout is avoided and the possibility to measure the transverse current also improves detection [85,170,175,176,177,178]. The flexibility and deformability of ultrathin membranes, however, have an impact on the translocation process. Since the confinement has a profound effect on the conformational entropy of a polymer, the shape of the channel becomes an important parameter. It has been found that an asymmetrical shape creates a gradient that drives the translocation process. This gradient thus reduces the threshold voltage required to facilitate the translocation [26,71,72,179,180]. Having a high voltage applied across the pore might result in false translocation events. Furthermore, as mentioned earlier, a cone-like shape only allows a few nucleotides around the one that is translocating due to the narrow constriction at the end [181,182], which enhances the accuracy of the current measurement.

The *pore–polymer interaction* is an important parameter; an attraction between the pore and the monomer enhances the capture rate, which in turn increases the number of successful translocation events. Furthermore, the translocation time depends on the strength of this interaction; for a moderate attraction, the translocation time decreases, whereas for a strong attraction, the polymer finds it difficult to leave the pore [160,180,183,184,185,186,187,188,189]. A polymer–pore interaction can be controlled by using tailor-made surfaces. Such surfaces can be achieved by either changing the composition of the membrane material [190,191] or by coating/fictionalizing the pore surfaces [192,193,194,195,196,197]. Rincon-Restrepo et al. proposed the use of molecular brakes, such as the one shown in Figure 6, to manipulate the translocation speed [197]. These molecular brakes are basically positive charges introduced in the lumen of the pore. Figure 6a shows a biological nanopore with a synthetic positive charge (red), whereas Figure 6b shows a hybrid one (a biological nanopore with a net positive charge on its surface is inserted into an SSNP). The speed of translocation can be controlled by the number of positive charges in the barrel. Moreover, in experiments, it has been shown that the polymer–pore interaction can be tuned by changing the pH gradient across the pores [198,199]. In Figure 7, we show the results obtained Joen and Mutthukumar, where it can be seen that the charge at the end of the β-barrel of the α-hemolysin pore varies as a function of the pH. For low pH values, the end of the pore is positively charged and attracts negatively charged molecules.

It has also been shown that the *location and the distribution of the interaction sites* inside the pore significantly affect the translocation time distribution [200,201,202,203,204,205,206]. Tuning the surface interaction patterns carefully may give tight control over the transport across the nanopore. Another way to modify the pore–polymer interaction is the pore morphology. It has been shown that translocation time can be increased by creating corrugations on the pore surface (see Figure 8 for an example of a corrugated pore). The corrugations result in a decreased effective pore diameter, an increased effective pore length, and a substantial increase in pore friction. Collectively, these factors lead to an enhanced translocation time [207,208,209,210,211].

In the biological setup, apart from the trans-membrane potentials, the translocation is also assisted by the binding proteins, called *chaperons* [212]. This mechanism is more common in protein translocation, but it is also relevant for the transport of DNA through membranes [213]. These chaperons are freely diffused binding proteins that bind to the monomers on the trans side and prevent any back-sliding of the polymer, giving rise to what is called Brownian ratcheting. Figure 9 shows the translocation of polymers in the presence of chaperons on the trans side of the pore; these results were obtained by Adhikari and Bhattacharya using Brownian Dynamic simulations [79]. Apart from creating a Brownian ratchet, these chaperons provide a driving force that depends on factors such as the binding rate and binding strength between the chaperon and the monomers, the type of bonding (e.g., univalent/ multivalent, binding reversibly or not, active/passive, etc.), and the concentration and distribution (e.g., size distribution among the chaperons and spatial distribution) of the chaperons [76,78,79,214,215,216]. The translocation dynamics also depend on polymer properties such as the flexibility, sequence, and length of the polymer [217,218,219]. Apart from chaperons, the cellular environment contains other macromolecules such as proteins and ribosomes. The volume fraction of large macromolecules in the cell interior could be as large as 50% [220]. The presence of these macromolecules affects the conformational and dynamical properties of the polymer, which in turn affects the translocation dynamics. The translocation dynamics can be controlled by changing the parameters, such as the size, volume fraction, activity, and distribution of these crowders [52,80,216,221,222,223,224,225,226,227]. This strategy comes with ease of use; it just requires adding crowding agents to the solution. Another manifestation of the crowding effect is in the use of obstacles on top of the nanopores [228,229,230].

The last proposed strategy that we would like to touch upon here is the *modification of the solvent properties*. This includes changing the electrolyte species, setting up a concentration gradient, and changing the viscosity or the temperature of the solvent. Counterion binding to the negatively charged DNA molecules reduces the charge and hence the electrophoretic force on DNA. The reduction in electrophoretic force, which in turn enhances the translocation time, depends upon the ionic species present in the electrolyte. It has been shown that the translocation time of DNA is longer in LiCl than in NaCl and KCl [138], as the binding of Li${}^{+}$ with DNA is atronger than that of Na${}^{+}$ and K${}^{+}$. Using counter ions of higher valency has further implications [231,232,233], as the valency of the ions in solution affects the conformation and charge distribution of the DNA molecule [234,235]. The translocation time also increases if the concentration of the counterions is increased in the solution [231], which, however, leads to a reduced capture rate. Creating an ionic gradient across the pore might remedy this issue and increase the capture rate; having a lower salt concentration on the *cis* side increases the capture rate without decreasing the translocation time [124,236,237,238,239,240]. Increasing the viscosity of the solvent again reduces the mobility of the DNA strands, resulting in longer translocation times. The capture rate can be increased by creating a viscosity gradient across the pore; the gradient basically gives rise to the pumping effect [47,50,60,137,241,242]. Changing the temperature of the solvent has similar effects, temperature can affect solution properties such as viscosity and conductivity and thus affects the translocation time. A temperature gradient across the pore, again, leads to an increase in the capture rate [89,243,244,245,246,247]. Another approach that we will quickly mention here makes the use of edge leaked field to trap the polymer in the pores [248,249].

## 5. Summary and Future Perspective

Polymer translocation through nanopores is at the core of various biological processes, such as viral DNA ejection and transport of proteins across membranes. The process is also relevant to various technological applications, developing low-cost and efficient bio-sensing devices being one of the most prominent. These sequencing devices work by measuring the modulation in ionic current when a molecule passes through the nanopore. Since the seminal work of Bezrukov et al. [250] and Kasianowicz et al. [8], there has been a wealth of research concerning polymer translocation through nanopores. This includes theoretical modelling and simulation [21,38,69,251], as well as experimental studies [9,38,188,252], with a focus on understanding the translocation behavior under varying conditions. This understanding is vital in developing cheap and efficient bio-sensors. When comparing the experimental and theoretical findings, the focus has been on the scaling behavior of the translocation time (τ) with system parameters such as the chain length. A wide range of scaling has been obtained depending upon the fine details of the system, and a unified theory of polymer translocation has yet to emerge. We have given a short review of the theoretical/simulation studies and obtained scaling behavior in different contexts in Section 2.

Having a suitable nanopore is crucial to the accuracy and efficiency of sequencing devices. Pores could be either of biological origin (transmembrane protein channels) or could be produced using thin films of materials such as graphene, SiN, or MoS${}_{2}$ (solid-state nanopores). The very first experiments on polymer translocation used biological pores such as α-hemolysin. These pores, however, come with their own limitations, such as their limited chemical and mechanical stability. Solid-state nanopores, on the other hand, are robust and their shapes and sizes can be tuned with reasonable control. Creating pores of a few nano-meters in films of subnanometer thickness, however, is tricky. If the thickness of the membrane is more than the distance between two nucleotides, the accuracy of detection is affected. Various techniques have been proposed to create ultra-small nanopores in ultra-thin membranes. A third approach combines elements from both the solid-state and biological nanopores, e.g, insertion of α-hemolysin into a solid-state nanopore. This hybrid approach gives solid-state nanopores a chemical specificity similar that of the biological ones. Details about these can be found in Section 3 and in the following review articles [9,10,11,21].

Despite the tremendous efforts and rich findings of the past, polymer translocation-based, low-cost, high-efficiency sequencing devices are yet to be realized. The key challenge lies in the control over the speed of the translocating polymer. As mentioned earlier, the polymer translocation does not follow any universal scaling law and certain details of the system may strongly influence the transport dynamics. Since the process is governed by a a complex interplay of a number of parameters, it is important to de-convolute their effects. In this spirit, a plethora of studies have investigated the effect of various system parameters on translocation dynamics, and a number of strategies to slow down or control the translocation have been proposed as outcomes of these studies. These strategies include changing the interaction between the pore and the polymer, changing the shape and the size of the pore, introducing chaperons and crowding agents, using obstacles near the pore, modifying the properties of the solvents, and so on. We have summarised most of these proposed remedies and limitations thereof in Section 4. Simulations play a critical role in accessing the effect of various parameters on the translocation dynamics and can be very useful in providing the optimal design parameters. Besides, with advances in experimental techniques, it is also possible to experimentally monitor the translocation dynamics at scales smaller than the chain length [108]. Such experiments, combined with simulations and theoretical modeling, would increase our understanding of the transport process, thus enabling better control over the process.

Though not very straightforward to implement, pulled translocation poses a potentially useful method for sequencing. It has been observed that pulling the polymer from one end using AFM or optical/magnetic tweezers gives tight control over the translocation dynamics. The speed could be manipulated and polymers could be even reverse translocated by choosing the right set of parameters. Another promising strategy is to modify the polymer–pore interactions; having an attractive pore–polymer interaction increases the capture rate, which also results in an increased number of successful events. One way of achieving this is by chemical modification of the pore surface. However, with better possibilities of creating functionalized nanopores, e.g, using DNA-nanoconstructs, it should be easier to tune the pore–polymer interactions. Chaperons/crowders are an integral part of the translocation process in physiological conditions and have been shown to have a significant impact on translocation dynamics. Besides, adding chaperons or crowding agents to the set-up is easy to implement. Combining more than one of the proposed strategies might lead to better control over the translocation. Moreover, experimentalists have come a long way; from using biological nanopores to creating solid-state nanopores with complex structures and great tunability. A great amount of understanding on the effect of various system parameters is already available from past studies. We believe that combining the insights and findings from various areas of research on polymer translocation, as well as sensors, would lead to the realization of practical, low-cost, high-efficiency genome sequencing devices.

## Figures and Tables

**Figure 2 ijms-24-06153-f002:**
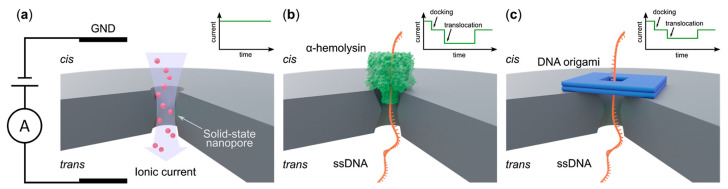
Types of nanopores: (**a**) the figure shows ion transport through a solid state nanopore fabricated on a substrate such as Si${}_{3}$N${}_{4}$, graphene, etc. The translocation of a single-stranded DNA is shown in (**b**) through a α—hemolysin pore, a natural/biological pore embedded in lipid bilayers, and (**c**) via a double-layer DNA origami nanopore, an example of hybrid nanopores. The inset shows the corresponding variation in recorded current during the translocation. This figure is reproduced from [10].

**Figure 3 ijms-24-06153-f003:**
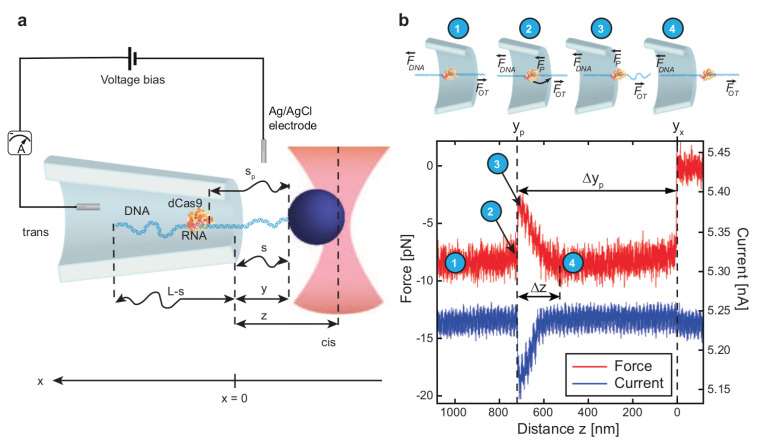
End—pulled translocation of a DNA—protein complex (DPC) through a nanopore using an optical tweezer. (**a**) shows a schematic diagram of a typical experimental set-up, where one end of a polymer is grafted on an optical bead and the motion of this bead is controlled using an optical tweezer (**b**). Cartoon representations of the different stages of translocation are shown in (b1–b4), and the signature of these stages can be traced out as a jump or traps in the recorded current/force vs. distance diagram (bottom panel of (**b**)). Reprinted with permission from [158] copyright 2016 by *American Chemical Society*.

**Figure 4 ijms-24-06153-f004:**
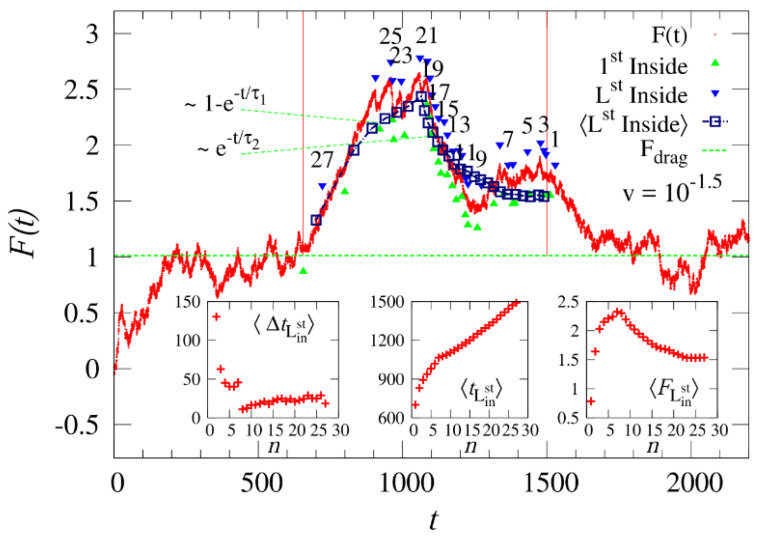
Force trajectory as a function of time for a polymer with 32 monomers. For other system parameters, kindly see the work of Fiasconaro and Falo [163]. The blue (above the curve) and green (below the curve) arrows indicate, respectively, the time of the last and first entrance of the monomer (indexed above the blue arrow) inside the pore. The squares indicate the average force at the entrance events (the last entrance in the case of multiple entrance exits). The horizontal corresponds to the value of the drag force $F_{drag}$ in the absence of pore-polymer interactions and fluctuations. The two (red) vertical lines indicate the time interval over which the mean force is calculated. The three insets correspond to the mean waiting entrance times, the total time spent to enter, and the mean force at the last entrance as a function of the number of monomers that enter the pore, respectively. Reprinted with permission from Fiasconaro and Falo [163] copyright 2018 by the *American Physical Society*.

**Figure 5 ijms-24-06153-f005:**
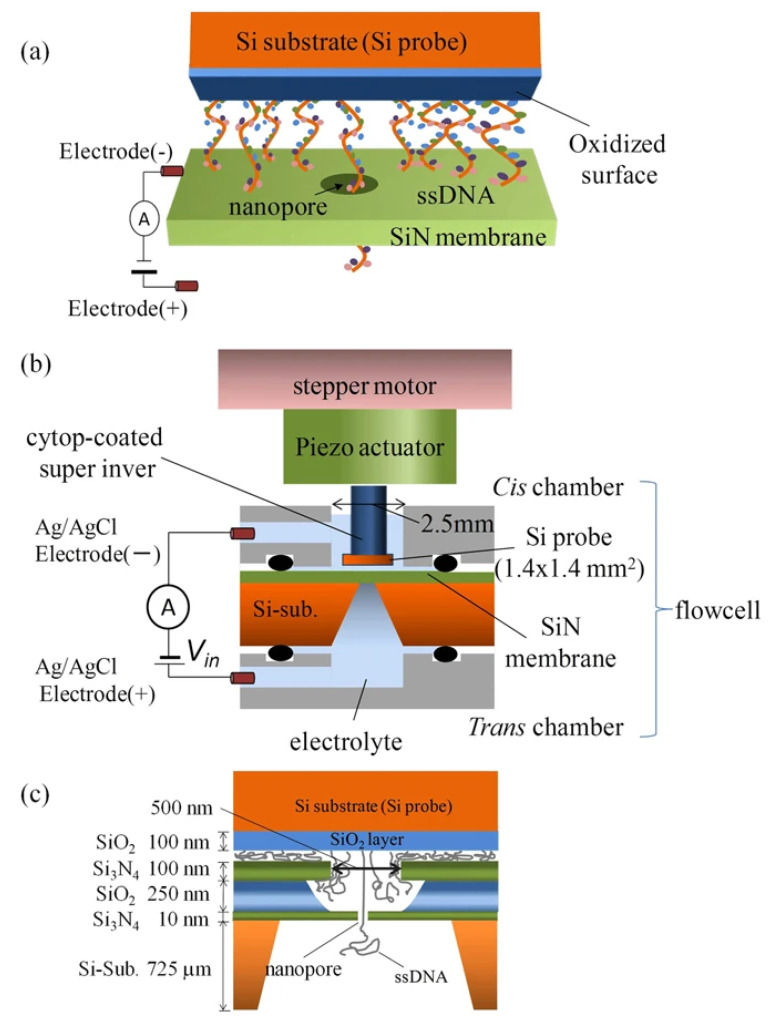
Schematic representation of immobilizing the DNA strands on a probe and using the system for sequencing purposes. (**a**) shows a cartoon of ionic current measurement when the probe remains in the nanopore. (**b**) Scheme of the measurement setup. (**c**) Zoomed-in schematic for around the nanopore. This scheme is reproduced from [165].

**Figure 6 ijms-24-06153-f006:**
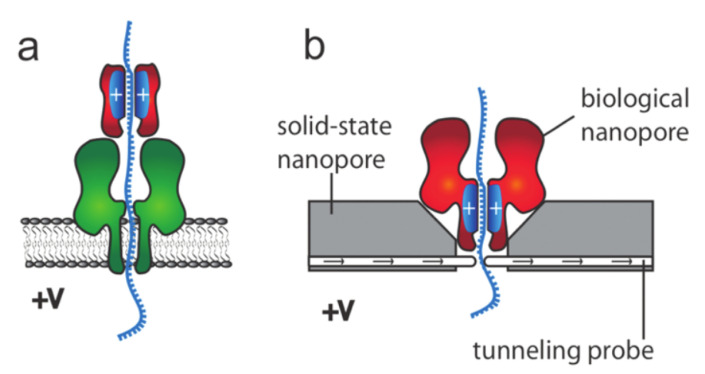
Nanopores with molecular brakes; (**a**) shows a protein nanopore with a synthetic protein barrel which is positively charged, whereas (**b**) shows a hybrid nanopore where the protein channel contains an internal positive charge. Reprinted with permission from [197] copyright 2016 by *American Chemical Society*.

**Figure 7 ijms-24-06153-f007:**
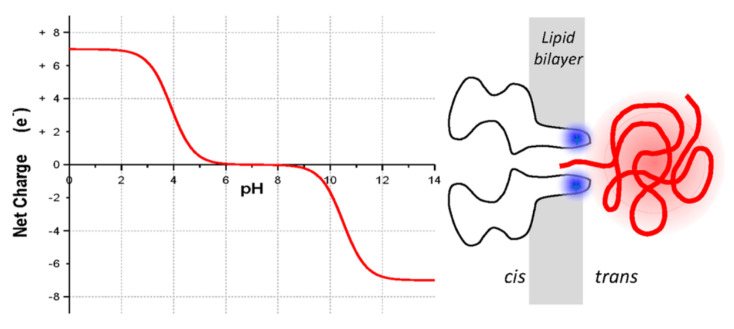
Results obtained by Jeon and Muthukumar [199] showing the net charge at the end of an α—hemolysine pore as a function of pH. Reprinted with permission from [199] copyright 2016 by *American Chemical Society*.

**Figure 8 ijms-24-06153-f008:**
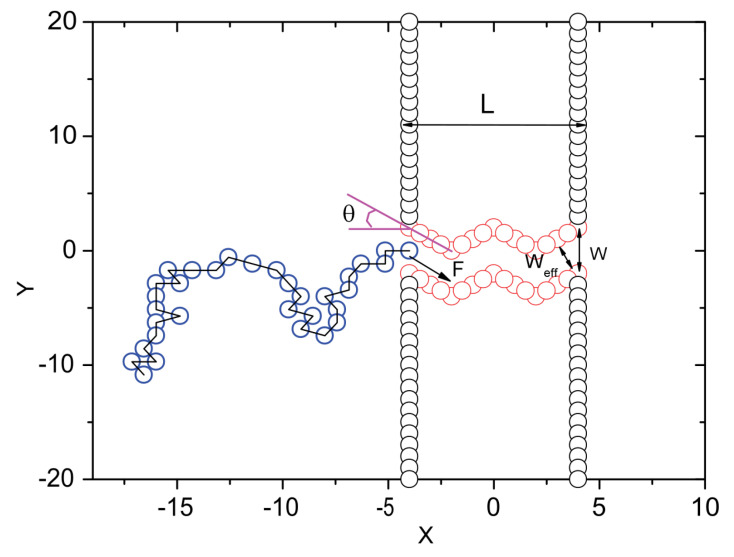
Scheme of a corrugated nanopore as proposed by Wang et al. [208], the corrugation is modeled via kinks in the pore. The translocation happens in the presence of an external driving force, *F*. Reprinted with permission from [208] copyright 2015 by *American Institute of Physics*.

**Figure 9 ijms-24-06153-f009:**
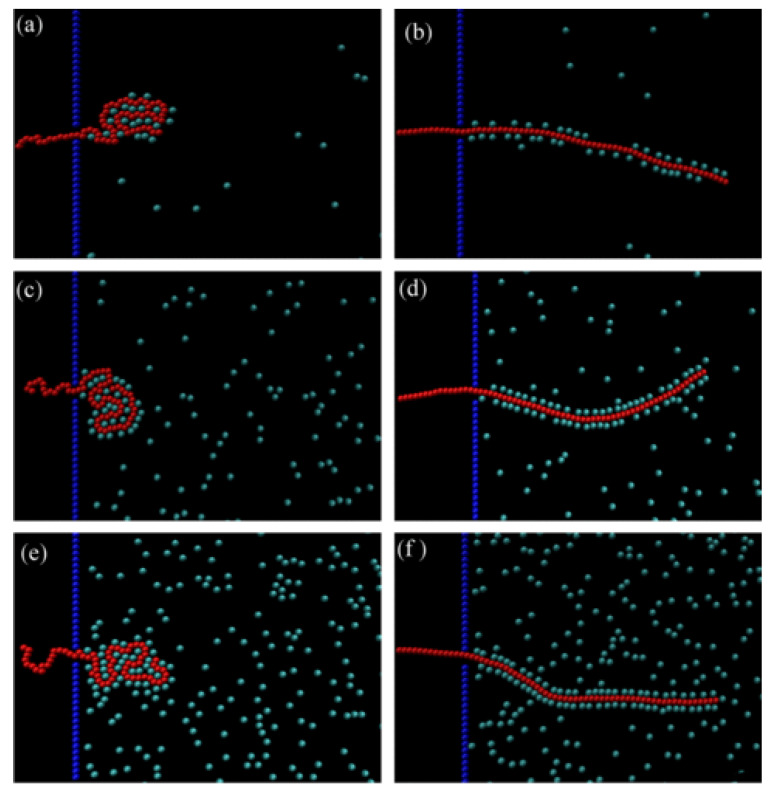
Translocation in the presence of chaperons, where the cyan circles represent the chaperons and the red circles represent the chain monomers. The left panel shows snapshots of the process for a fully flexible chain, whereas the right panel shows the same for a very stiff chain. The studies were performed using Brownian dynamic simulations for three chaperons densities for both chains. (**a**,**c**,**e**) correspond to three densities, ρ = 1%, 5%, and 10%, respectively, for a fully flexible chain, whereas (**b**,**d**,**f**) correspond to a stiff chain (κ=256) for the same densities. This figure is reprinted with permission from [79]. Copyright 2015 by the *American Physical Society*.

## Data Availability

Not applicable.

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
