# Peer review of "Polymer Translocation and Nanopore Sequencing: A Review of Advances and Challenges"

_ijms, 2023, doi:10.3390/ijms24076153_

Round 1

Reviewer 1 Report

The topic of this review is certainly of interest. However, in the present form it's hardly satisfactory and need a lot of work.

The main problem with the review: it seems pointless. The entire discussion of the simulation work leads us almost nowhere. The conclusion is very general and weak. DNA deciphering (this is what the authors discuss for the most part) is a very mature field with benchmarks and competing techniques. Do the authors think the translocation-based approaches will become competitive and what exactly achievements (or failures) should make us think they will (or will not) compete. What about the quality of deciphering (hardly discussed) vs speed? This is a review on *technology* (even potential) and should be written as such.

On discussion of scaling exponents and anomalous mobility: there are reports that $1+\nu$ only holds for reasonably good solvent, which suggests it's just a number (somewhat close to 1.5 or 1.57) rather that a dependence, because $1+\nu$ does not hold when the solvent quality worsens (eg Yang et al 10.1021/jp3104672 but there are more) . Also, there are several papers by D. Talaga, where he fitted the translocation time distribution by the regular (not fractional) Fokker-Plank eq, which means non-anomalous mobility of the polymer.

There are several generally incoherent subsections , where the authors jump back-and-forth in their discussion.

I really wonder whether the authors know examples of translocation as a fast approximate method for non-specific analysis of polymers (estimating general characteristics such as length, copolymer composition, homogeneity etc). Not really an explored topic, would be nice to know.

Finally, does the paper fits to JIMS scope? Most of the discussion is structured around not-too-specific topics of polymer physics, which is not the focus of JIMS. Among MDPI journals, Polymers would be a reasonable fit.

The topic is good but the paper needs A LOT of work

minor comments

there is no need to start "chaperones" with a cap

"realized" is misused in several instances

the authors spend to much effort in each section describing how they will describe the material, which is just a symptom of poor structuring of the text

Reviewer 2 Report

This a well-written review-type article which I am happy to recommmend for publication in IIJMS.

Reviewer 3 Report

The authors review polymer translocation and nano-pore sequencing. I believe that this manuscript can be considered for publication in this Journal after minor revisions.

 - authors should provide a complete table, including the reviewed studies, in the manuscript.

Round 2

Reviewer 1 Report

Unfortunately, in the present form this is a poor quality review that should not be published. A review "easy to read for beginners " (as the authors say) should exactly start with the place of the technology on the current landscape. For example, if the translocation based techniques have no whatsoever chance compete with NGS, do one even needs to bother reading such a review? Right now translocation is not an established technology, it is far behind. The discussions whether solid-state pores have any prospects for such a task or we should deal with membrane proteins only was pretty much active around 2014, nine (!) years ago. Has anything crucial (for technological applicability of the idea) happened since then? 

Since the authors review a prospective technology that has fallen behind, the sense in such a review is identifying the gaps that should be bridged across to make it work. Right now the review can be understood exactly by those who has been working in the field for a while, and does not say much to the outsiders, which is dramatically lowers the quality of the work. Adding such a page should be much easier than discussing the scaling exponents, to be honest.

As for the work of David Talaga: there is no implication that the mobility constant should be the same for all polymer length. But if the diffusion is anomalous, the distribution of translocation times should not be reasonably described by the FP equation. So basically, since for every single length the distribution is described by the FP eq., the diffusion is not anomalous. Otherwise one needs to employ the fractional FP eq. This follows from the very derivation of the FP equation. As a theoretician, I simply cannot understand the pages written on theoretical & simulation studies and ignoring the experimental evidence.

Reference to Yang et al is missing (although the name is mentioned)

Author Response

Referee’s remark:

Unfortunately, in the present form this is a poor quality review that should not be published. A

review "easy to read for beginners " (as the authors say) should exactly start with the place of the technology on the current landscape. For example, if the translocation based techniques have no whatsoever chance compete with NGS, do one even needs to bother reading such a review? Right now translocation is not an established technology, it is far behind. The discussions whether solid-state pores have any prospects for such a task or we should deal with membrane proteins only was pretty much active around 2014, nine (!) years ago. Has anything crucial (for technological applicability of the idea) happened since then?

Our response: Unfortunately, we are misquoted here: Easy to read reviews are important (writing such reviews is quite difficult given that the subject has to be made easy to understand along with stimulating readers’ interest in the field), beginners in the field are equally important given that they will be contributing for next 25-30 years. Besides, all the relevant references are given here, if someone can explore their interests, they can get advanced information by following the given references. That's the purpose of our review. 

Regarding where should we start with; we have said before also, this is not a review about various existing technologies for NGS, it is a review about polymer translocation based sequencing devices and everything is written in that framework.

Referee’s comment(part of the first comment):

``For example, if the translocation based techniques have no whatsoever chance compete with NGS, do one even needs to bother reading such a review? Right now translocation is not an established technology, it is far behind.”

Our response: This is a personal belief of our esteemed referee.  We believe that translocation based sequencing has the potential to provide a label-free and cost-effective method for sequencing,  and so does a sizeable community of researchers actively working

on understanding the polymer translocation dynamics. This is easily manifested in the number of related papers published every month (one can easily find on google scholar, or we can provide the links). Now, as much as we are free to believe in whatever we like, we have to respect what others are doing!

As pointed out by the referee, translocation based sequencing has not established yet (work under progress), more research is required and, thus, our review becomes more relevant!

Referee’s comment (last part of the first remark):

The discussions whether solid-state pores have any prospects for such a task or we should deal with membrane proteins only were pretty much active around 2014, nine (!) years ago. Has anything crucial (for technological applicability of the idea) happened since then?

Our Response: Since it is a review, we have given an overview of whatever has been done using SSNPs, we have also summarized the studies, and results thereof, involving biological and hybrid nanopores. It is a review article, it has to list things that have been done in the past, and SSNPs have been actively studied and are still being studied.  If at all, this is what we write in future prospects (Last section):

“However, with better possibilities of creating functionalized nano-pore, e.g, using DNA-nanoconstructs, it should be easier to tune the pore-polymer interactions.”

Referees remark: Since the authors review a prospective technology that has fallen behind, the sense in such a review is identifying the gaps that should be bridged across to make it work. Right now the review can be understood exactly by those who has been working in the field for a while, and does not say much to the outsiders, which is dramatically lowers the quality of the work. Adding such a page should be much easier than discussing the scaling exponents, to be honest.

Our Response: For us, it is not a technology that has fallen behind, but a work in progress and

simulations play an important role in providing system parameters, we feel that it is important to give the readers a concise review of theoretical/simulation studies.

Referee’s remark:

As for the work of David Talaga: there is no implication that the mobility constant should be the same for all polymer length. But if the diffusion is anomalous, the distribution of translocation times should not be reasonably described by the FP equation. So basically, since for every single length the distribution is described by the FP eq., the diffusion is not anomalous. Otherwise one needs to employ the fractional FP eq. This follows from the very derivation of the FP equation. As a theoretician, I simply cannot understand the pages written on theoretical & simulation studies and ignoring the experimental evidence.

Our Response:  Again, it is a review article, we have tried to collect all the important studies including the ones that conclude that the dynamics is anomalous! Those are the peer-reviewed papers written by well established theoreticians. We, again, are not pronouncing any judgment here about the nature of the translocation dynamics, we are just reporting everything that has been done in the past. 

Referee’s remark:

Reference to Yang et al is missing (although the name is mentioned)

Our Response:  We apologize and we added the reference mentioned

Round 3

Reviewer 1 Report

Expressing my opinion on a particular problem is not a part of the review process. However, translocation based techniques are behind. It's a fact, not an opinion: there are commercially available techniques. I am simply asking authors to explain significance of their work: why the topic (translocation-based DNA deciphering) is significant. The critical review needs a real conclusion. I can repeat: what gaps in fundamental and practical sense should be bridged across, what are the hopes that the gaps will be bridged? I am not expressing any personal opinion on the prospect of the technology, I asking the authors to formulate the outcome of the review. 

An explanation of significance increases the number of readers of a paper two-fold at least.

As for the abovementioned ref of D.Talaga (I am not an author of that work!), a single measurement is more of significance to the field than many papers of respected theoreticians. The authors do not need to make a conclusion on the anomalous transport (true), but if they do venture into the discussion of the issue, they should discuss the experimental evidence on both sides. 

I am not willing to review this paper again.